# Exposure to Particulate PAHs on Potential Genotoxicity and Cancer Risk among School Children Living Near the Petrochemical Industry

**DOI:** 10.3390/ijerph18052575

**Published:** 2021-03-04

**Authors:** Nor Ashikin Sopian, Juliana Jalaludin, Suhaili Abu Bakar, Titi Rahmawati Hamedon, Mohd Talib Latif

**Affiliations:** 1Department of Environmental and Occupational Health, Faculty of Medicine and Health Sciences, Universiti Putra Malaysia, UPM Serdang 43400, Selangor, Malaysia; norashikinsopian@gmail.com; 2Department of Biomedical Sciences, Faculty of Medicine and Health Sciences, Universiti Putra Malaysia, UPM Serdang 43400, Selangor, Malaysia; suhaili_ab@upm.edu.my; 3Department of Community Health, Faculty of Medicine and Health Sciences, Universiti Putra Malaysia, UPM Serdang 43400, Selangor, Malaysia; rahmawati@upm.edu.my; 4Department of Earth Sciences and Environment, Faculty of Science and Technology, Universiti Kebangsaan Malaysia, Bangi 43600, Selangor, Malaysia; talib@ukm.edu.my

**Keywords:** polycyclic aromatic hydrocarbons (PAHs), children, DNA damage, industry

## Abstract

This study aimed to assess the association of exposure to particle-bound (PM_2.5_) polycyclic aromatic hydrocarbons (PAHs) with potential genotoxicity and cancer risk among children living near the petrochemical industry and comparative populations in Malaysia. PM_2.5_ samples were collected using a low-volume sampler for 24 h at three primary schools located within 5 km of the industrial area and three comparative schools more than 20 km away from any industrial activity. A gas chromatography–mass spectrometer was used to determine the analysis of 16 United States Environmental Protection Agency (USEPA) priority PAHs. A total of 205 children were randomly selected to assess the DNA damage in buccal cells, employing the comet assay. Total PAHs measured in exposed and comparative schools varied, respectively, from 61.60 to 64.64 ng m^−3^ and from 5.93 to 35.06 ng m^−3^. The PAH emission in exposed schools was contributed mainly by traffic and industrial emissions, dependent on the source apportionment. The 95th percentiles of the incremental lifetime cancer risk estimated using Monte Carlo simulation revealed that the inhalation risk for the exposed children and comparative populations was 2.22 × 10^−6^ and 2.95 × 10^−7^, respectively. The degree of DNA injury was substantially more severe among the exposed children relative to the comparative community. This study reveals that higher exposure to PAHs increases the risk of genotoxic effects and cancer among children.

## 1. Introduction

Particle-bound polycyclic aromatic hydrocarbons (PAHs) are highly lipophilic and pervasive harmful organic pollutants. They are present ubiquitously in the environment and eventually enter the human body through three main routes: inhalation, ingestion and dermal absorption [1,2,3]. They can be absorbed into air particles, with carcinogenic and mutagenic properties [4,5,6,7]. The Agency for Toxic Substances and Disease Registry (ATSDR) ranks PAH congeners at 9 of 275 in a priority list of hazardous substances due to the potential severe health threat for humans and the environment [8].

Atmospheric PAHs, primarily derived from anthropogenic activities such as industrial operations, vehicle exhausts, refineries, waste incineration and domestic heating, are emitted by incomplete combustion of organic matters at high temperatures [9]. Numerous international studies in petrochemical industrial areas and oil refineries have tracked particulate contamination and severe particulate PAH emissions [10,11,12,13,14]. Nevertheless, the adverse health effects on children living in nearby industrial areas have not been extensively investigated.

Owing to their developmental stage and physical and biological conditions, children are among the most vulnerable populations [15]. They are known to have a low metabolism capacity than adults; therefore, contaminated air inhalation can increase the metabolism burden in their small bodies [16]. Health symptoms encountered early in life increase a child’s future risk of disease and lead to permanent consequences [15]. In previous research, PAH exposures among a vulnerable population, such as children, have been highlighted [16,17,18,19,20,21,22,23]. Inhalation of particulate-bound PAHs was found to have a significant impact on the well-being of children and to be correlated with their growth [24], lung function impairment [25], obesity [26], low-grade inflammation [2], respiratory symptoms, tachycardia and cell damage [27].

Exposure to high urban and industrial air contaminant concentrations has been closely correlated with genetic damage in the children [20,28,29,30]. Research by Sánchez-Guerra et al. found that school children living in industrial areas and heavy traffic suffered significant DNA damage compared to students living in less polluted areas [31]. Investigations of the chronic impact of exposure to PAHs helped understand the contribution of environmental exposure as a risk factor for cancer, especially in moderate or low air pollution, with biomarker application [32,33,34]. In addition, increasing proximity to the industrial zone and the busiest road exposed children to higher PAH contamination levels, increase the risk of childhood cancer and respiratory implications [35,36,37,38,39].

This present study aims to quantify the level of particulate PAH exposure in the indoor and outdoor environments of schools located in the vicinity of the petrochemical industrial area. The exposure sources of PAHs were identified based on the source diagnostic and principal component analysis (PCA). The incremental lifetime cancer risk (ILCR) for children due to PAH inhalation was assessed using Monte Carlo simulation. Moreover, this study investigated the association of exposure to particulate PAHs with potential genotoxicity among primary school children, as indicated by the DNA damage level in their buccal epithelial cells.

## 2. Methods

### 2.1. Study Location

The 24 h PM_2.5_ (fine particle with a diameter generally less than 2.5 µm) sampling for five consecutive school days was carried out at three primary schools located within 5 km of the petrochemical industrial area, which were referred to as schools S1, S2 and S3. Meanwhile, schools C1, C2 and C3 were three comparative schools situated more than 20 km away from the industrial area and away from any industrial activity. Both the exposed and the comparative regions were situated in Terengganu, Malaysia (Figure 1). A comprehensive description of the sampling sites can be found in Appendix A. The PM_2.5_ samples were collected between January to May 2017.

### 2.2. Study Population

The comparative cross-sectional analysis included 234 children aged between 9 and 11 years who met the inclusion and exclusion requirements of the respondent selection. These criteria were determined to evaluate the confounding factor because there was a significant association between the increase in chromosomal damage in a respondent and chronic diseases [40]. Children with a family history of cancer who received radiotherapy or chemotherapy in the past 12 months or X-rays in the past 3 months were excluded from this research. Any radiation exposure might have induced cytogenetic alteration and influence the validity of the outcome of this study [41]. Before the sample collection was conducted, written informed consent was sought from the children’ parents or guardians.

### 2.3. Measurements of Particulate Matter 

A total of 60 indoor and outdoor PM_2.5_ samples were collected using a low-volume air sampler equipped with cyclone (Airmetrics Mini Vol, Springfield, OH, USA) with a flow rate of 5 L min^−1^ through weighted quartz microfiber filters (47 mm diameter). The air sampler was positioned approximately 1 m above the floor and was placed in the school garden and the preferred classroom in a safe area.

### 2.4. PAH Analysis

With a few modifications, the extraction of particulate PAHs by Sulong et al. and Khan et al. was adopted [5,42]. In a glass bottle containing a mix of 10 mL of dichloromethane (DCM) and n-hexane (5 mL:5 mL), filter paper was cut into small pieces. For a total of 30 min, the sample was sonicated in a bath sonicator, with a 2 min run and 1 min rest. A filtration unit containing an annealed glass fibre filter was used to filter any residual insoluble particles, and the collected solution was purified. Under a gentle blow of nitrogen gas, the extract was concentrated to approximately 0.2 mL of the extract. Next, to reconstitute the extract residue, 0.7 mL of n-hexane was added gently and spiked with 0.1 mL of 4.0 ppm standard surrogate perylene before transferring it into a cartridge for solid-phase extraction (SPE).

To clean up and pre-concentrate samples, a silica SPE cartridge (Agela Cleanert CN-SPE 1000 mg 6 mL^−1^) was used. The SPE cartridge was conditioned with 10 mL of n-hexane to activate the packing material of the chromatographic sorbent and to allow a proper phase interface with the sample. The final procedure of the SPE operation was followed by PAH elution by a mixture of dichloromethane and n-hexane (3.5 mL:6.5 mL). Under a gentle stream of nitrogen gas, the eluent was blown down until scorched. Before transferring to a 2 mL autosampler vial containing a vial insert, it was diluted with 0.1 mL of n-hexane.

The gas chromatography–mass spectrometry (GCMS) instrument (Agilent Technologies) was calibrated with standard mixtures of PAHs such as naphthalene (NAP), acenaphthene (ACP), acenaphthylene (ACY), anthracene (ANT), fluorene (FLU), phenanthrene (PHE), anthracene (ANT), fluoranthene (FLA), pyrene (PYR), benzo(a)anthracene (BaA), chrysene (CYR), benzo(b)fluoranthene (BbF), benzo(k)fluoranthene (BkF), benzo(a)pyrene (BaP), indeno(1,2,3-cd)pyrene (IcP), dibenzo(a,h)anthracene (DbA) and benzo(ghi)perylene (BgP). The GC column temperature was programmed as follows: initial 40 °C, followed by a temperature increase to 150 °C (8 °C per min), and an increment of 4 °C per min until 310 °C for a 6 min hold. The PAH concentration was determined through an internal calibration standard containing a known concentration mixture of 16 PAH congeners. The final concentration PAHs in the PM_2.5_ was calculated using the following equation, in which the total air volume was 7.2 m^3^:C (ng m^−3^) = C determined (ng ml^−1^) × dilution factor/air volume (m^3^)(1)

### 2.5. Health Risk Assessment

Risk assessment is an essential method for assessing the adverse effects of PAH exposure on human health, defining the target organ and identifying the particular effects of chemicals to find an appropriate risk minimisation solution [43]. The carcinogenicity potency PAHs can be determined based on the benzo(a)pyrene equivalent concentration (BaPeq), also known as toxicity equivalent concentration (TEQ). Obtaining the TEQ for the PAH compounds requires the involvement of reference toxicity equivalent factors (TEFs) proposed by Nisbet and Lagoy [44], relative to the carcinogenic potency of BaP. The TEQ was determined based on a constant TEF value multiplied with the individual PAHs’ concentration, as indicated by the following equation:TEQ = 0.001 (NAP + ACP + ACY + FLU + PHE + PYR) + 0.01 (ANT + BgP + CYR) + 0.1 (BaA + BbF + BkF + IcP) + BaP + DbA(2)

Afterwards, the incremental lifetime cancer risk (ILCR) model was used to estimate the carcinogenic risk due to respiratory exposure to PAHs [45,46,47]. The following equation was employed to calculate the unitless ILCR of PAH exposure in children:(3)$$\mathrm{ILCR}=\frac{C\;\times \sqrt[{3}]{\frac{\mathrm{BW}}{70}}\times \;\mathrm{IR}\;\times \;\mathrm{ED}\;\times \mathrm{EF}}{\mathrm{BW}\times \mathrm{AT}\;}\;\times \;\mathrm{CSF}$$
where C indicates the carcinogenic PAHs based on BaPeq (ng m^−3^) and IR is the air inhalation rate for children (12 m^3^ day^−1^) [48]. EF stands for exposure frequency or total schooling day in the year 2017 (250 day year^−1^), and ED denotes the exposure duration or length of time of contaminant contact for children (6 years) [48]. BW is the measured body weight of children during the sampling, and AT is the averaging time of carcinogenic PAHs exposure (70 years × 365 days) [48]. CSF refers to inhalation carcinogenic slope for BaP, which is 3.85 mg kg^−1^ day^−1^ [49].

The ILCR model prediction could either overestimate or underestimate the actual carcinogenic risk. Consequently, the outcome of cancer risk was not applied to the entire population of interest. Integrating variability and uncertainty into the risk model is a more precise approach to determine the probabilistic carcinogenic health risk. Therefore, Monte Carlo simulation with 10,000 iterations was employed in this study to appraise the probabilistic ILCR range of children exposed to PAHs for the exposed and comparative schools. The simulation also integrated the uncertainty and variability of the measured PAH concentrations and the specific determinant of children’s exposure [47,50,51]. Afterwards, sensitivity analysis was conducted to identify input parameters that significantly influence the estimated risk. Both analyses were implemented using Crystal Ball software (version 11.1.2.4; Oracle Corp., Austin, TX, USA). Table 1 portrays the variable distribution types used in the Monte Carlo simulation for both exposed and comparative schools.

### 2.6. Collection of Exfoliated Buccal Mucosa and MN Assay

Buccal mucosa cells were obtained by gently scraping the inner sides of both cheeks with a sterile cytology brush. The brush was then dipped into a microcentrifuge tube containing 0.1 M phosphate buffer solution (pH 7.5) and stored in a freezer at −20 °C. 

### 2.7. Comet Assay

The standard procedure of the Comet Assay Kit (Trevigen, Gaithersburg, MD, USA) was followed to conduct this assay. The cell suspension was washed with sterile phosphate-buffered saline (PBS) solution and centrifuged for 1 min at 2500 rpm to obtain the precipitate cells. The cells were mixed with 75 μL of molten low melting agarose (LMA) (Trevigen, Gaithersburg, MD, USA) and immediately embedded in the comet slide. Next, at 4 °C, the fixed cells underwent a lysing procedure in a pre-chilled lysis solution for 60 min, followed by immersion at room temperature for 60 min in a freshly prepared alkaline solution. At a constant voltage of 20 V, the slide was immersed for 20 min in an electrophoresis buffer. It was gently rinsed with 70% ethanol and deionised water after electrophoresis. The comet images were visualised with a 50 μL SyBr Green diluted solution staining effect. A fluorescence microscope randomly captured 100 comet images under 20× magnification (Motic, Hong Kong, China). Using OpenComet software, which has greater precision and can minimise human bias, the DNA damage was measured as a tail moment [52].

### 2.8. Statistical Analysis 

The Statistical Package for Social Science (SPSS) version 22.0 analysed all the data. The normality test determined the subsequent statistical analysis, either parametric or non-parametric tests. Bivariate analysis was applied for the mean comparison and the association of the parameters tested in this study. Multiple linear regression was performed to assess the factors contributing to DNA damage among the children.

Using XLSTAT 2020.5.1 software to classify and check the various sources of PAH pollutants in the sampling areas, principal components analysis (PCA) with varimax rotation was carried out [53]. The missing data were substituted by half of the value of the method detection limit (MDL) [18] prior to running PCA. Before proceeding with compatibility data testing, the entire dataset was converted to a dimensionless form. As far as ensuring the compatibility of PCA data was concerned, the Kaiser–Meyer–Olkin (KMO) test and Bartlett’s test of sphericity were performed [45]. A KMO value greater than 0.5 indicated that the PAH dataset was in the best condition to be applied in the PCA software.

Additionally, a significant chi-square of Bartlett’s test also indicated that the dataset was appropriate to be used. Using the varimax method with Kaiser standardisation, the major elements with eigenvalues greater than one were extracted and rotated. The PCA factor loadings were interpreted depending on the correlation value, which was assumed to be a heavy loading value above 0.75. Moderate loading varied between 0.50 and 0.75, and weak loading ranged from 0.30 to 0.50 [54].

### 2.9. Quality Control

To ensure the equilibrium of mass concentration, blank filters were conditioned in a desiccator for 24 h. All the glassware was first washed with running tap water and rinsed with distilled water, followed by hexane, methanol and acetone, to analyse PAHs. To volatilise and eliminate organic contaminants, the glassware and filter paper were baked in a furnace for 4 h at 400 °C [5,42].

All samples were spiked with 0.1 mL of 4 ppm internal standards (chrysene-D12 and perylene-D12) during sample preparation to test for potential organic contamination. For the internal standards, the recovery efficiency ranged from 76% to 95%. With minimal indirect light exposure to avoid photodegradation, all sample extraction procedures were carried out under a fume hood. Blank filters for the actual samples were extracted and analysed using the same approach to ensure no major background interference. To create calibration curves, five points (0.1 to 1.0 mg L^−1^) of the certified reference standard of the EPA 610 Polynuclear Aromatic Hydrocarbons Mixture (Supelco, Bellefonte, PA, USA) were analysed. The limits of detection (LOD) was determined on the basis of independent measurements of the blank sample and its standard deviation. The correlation coefficient and LOD of 16 USEPA priority PAHs are presented in Appendix A

Furthermore, the quality control for the comet assay included observation of the zigzag method to prevent overlapping views. The sample of buccal epithelial cells was properly shielded from direct sunlight. Moreover, sample preparation and examination were performed under dim light. During the scoring stage, a potential error in determining DNA damage may occur. To manually mitigate false-positive occurrences, OpenComet software used in the comet assay allows the user to discard the irregular comet shape. Interestingly, a statistical outlier analysis of each comet image was also applied to this software. The inconsistent image has been identified as an outlier and can minimise the number of false positives [52].

## 3. Results and Discussion 

### 3.1. Distributions of PAH Species at the Exposed and Comparative Schools

Figure 2 portrays the distributions of particulate PAHs in the exposed and comparative schools, sampled 24 h during school days. The highest concentration of total PAHs was recorded in exposed school S2, with a mean value of 67.72 ± 49.84 ng m^−3^, closely followed by school S1 (64.64 ± 44.85 ng m^−3^) and school S3 (61.60 ± 39.74 ng m^−3^) (Appendix A). On the other hand, the comparative schools located more than 20 km away from the petrochemical industrial areas reported low PAH concentrations, especially schools C2 (5.93 ± 0.59 ng m^−3^ and C3 (6.36 ± 1.19 ng m^−3^). School C1 demonstrated a contrary finding, with a concentration six times higher (35.06 ± 9.71 ng m^−3^) than the other two schools (5.93 ± 0.59 and 6.36 ± 1.19 ng m^−3^). The concentration trends in indoor PAHs for the exposed schools were as follows: S1 (63.22 ± 33.95 ng m^−3^) > S2 (54.97 ± 48.94 ng m^−3^) > S3 (44.27 ± 28.40 ng m^−3^). Meanwhile, comparative school C1 demonstrated the highest indoor PAH concentration (13.09 ± 8.93 ng m^−3^), followed by schools C2 (4.65 ± 0.77 ng m^−3^) and C3 (4.24 ± 1.14 ng m^−3^).

In general, the distribution of individual PAHs was strongly dominated by the species of heavy-molecular-weight (HMW) PAHs, structured by four to six rings (Appendix A). Schools C2 and C3 showed a contrasting finding; however, the low-molecular-weight (LMW) PAHs dominated the total concentration of PAHs relative to HMW PAHs. The outdoor LMW PAH concentrations of the three schools were 5.15 ± 0.52, 5.19 ± 1.16 and 16.38 ± 11.67 ng m^−3^, respectively. The findings also depicted that the ACY and ACP were barely undetected by the instrument due to the low concentration present in all samples. The majority of HMW organisms, especially in schools C2 and C3, were present in small and intense concentrations. BaP and PYR were the most prominent species in particulate PAH samples for all the exposed schools, with concentrations exceeding 10.00 ng m^−3^. Besides, PYR was also the highest species found in the outdoor samples of school C1. The higher fraction of HMW PAHs could be influenced by the degree of solvent extraction efficiency [55]. In this study, dichloromethane and n-hexane had similar polarity properties as the HMW PAHs, thus favouring the excellent extraction efficiency of the heavier PAH component.

On the other hand, particulate PAHs measured at schools C2 and C3 were highly influenced by FLU species in outdoor and indoor samples, with a concentration of 4.00 ng m^−3^. The results specifically demonstrated that schools within a distance of 5 km from the industrial areas were subject to higher PAH concentrations relative to comparative schools. In the outdoor samples, the results also revealed a higher degree of PAHs than the indoor samples. Traffic emissions, idling vehicles during school dismissals and parking could lead to this condition [56]. In addition, the location of air-sampling pumps in the shielded environment in the classroom (i.e., doors and windows were closed during school hours) may be the justification for the low concentration of indoor PAHs [57]. 

The PAH concentration in the research area was higher than in a study by Di Gilio et al., as the indoor concentration was up to 2.36 ng m^−3^ at a school near the steel industrial area [58]. BbF, BgP, IcP, BkF, and DbA species were identified as the most abundant constituents in that study, which was scientifically known as a proxy for traffic and coke oven emissions. Meanwhile, at the two primary schools in North Portugal, the industrial air pollutant and traffic exposure caused elevated PAH levels, with values of 20.00 and 48.00 ng m^−3^ [59]. In a school dormitory in an urban area of Tehran, Hassanvand et al. reported higher indoor and outdoor PAH concentration levels of 281.25 and 361.75 ng m^−3^, respectively [60]. The researchers eloquently mentioned that PHE, FLU, BaA, CYR and BgP were the most abundant PAH species, contributing significantly to fuel combustion activities (traffic and domestic heating emissions). Meanwhile, a school in Beijing demonstrated a higher PAH concentration (36.83 ng m^−3^) than the comparative schools, heavily dominated by ACY and FLA species [61]. 

Several subsequent studies on petrochemical air pollutant exposure have reported a higher PAH concentration than this present study. In a township within 10 km of the largest petrochemical complex in Taiwan, research by Yuan et al. found a higher PAH concentration, with an average PAH concentration of 15.20 ± 15.18 μg g^−1^ [11]. BbF, FLA, PYR, BaP, BgP and IcP were the most prominent PAH species and could be emitted from oil refinery plants and coal power plants. The concentration of particulate PAHs (PM_10_) from petrochemical-related factories in the Niger Delta, Nigeria, meanwhile, was 9.2 μg m^−3^ [62]. Similarly, in a petrochemical industrial complex in Ulsan, Korea, the PAH concentration obtained in road dust was 55.33 ± 15.83 μg g^−1^, which was comparatively higher than the current study [13]. The study eloquently defined that the ring number distribution of PAHs in the petrochemical area was close to a heavily trafficked area, possibly because of the similarity in pollution sources. A study by Bozlaker et al. in Turkey, however, showed that the comparable concentration of PAHs (particulate and gas) varied in summer from 7.30 to 44.80 ng m^−3^ [14].

### 3.2. Source Diagnostic Ratio

The diagnostic ratio of the analysis source determined the possible sources by contrasting parent PAH ratios with frequently found PAH emissions (Appendix A). The ratio of LMW/HMW PAHs was shown by the anthropogenic source apportion index, with values lower than 1 indicating pyrogenic emissions [63]. The ANT/(ANT + PHE) ratio was also an anthropogenic source indicator for PAH emissions, with values less than 0.1 suggesting a petrogenic source. In contrast, any value greater than 0.1 implies a pyrogenic source [64]. Pyrogenic sources were possibly correlated with around 89.5% (*n* = 17) samples from the exposed school. The petrogenic source was an IcP/(Icp + BgP) ratio of less than 0.2, and the fuel combustion source is denoted by a value ranging from 0.2 to 0.5 [65].

A diagnostic ratio value that is higher than 0.5 indicates combustion of grass, wood or coal. The data specifically showed that all classes, except comparative school C1, which had a mean ratio of 0.46, were predominantly grass-, wood- or coal-combustion-sourced. Relatively, exposed school S1 had the highest sample number (*n* = 7, 63.6%) originating from fuel combustion.

In comparison, coal combustion and vehicular emissions can be distinguished by BaA/CYR [65]. A value of 0.2 and 0.35 denoted coal combustion, while vehicular emissions were responsible for a value greater than 0.35. The ratio indicated that the concentration of PAHs could have been caused by coal combustion, with 10 of the exposed school samples providing a large pollution source tracer. However, the majority of the samples were predominantly from traffic pollution. A diagnostic ratio of BaA/(BaA + CYR) was proposed by Yunker et al. for the determination of the PAH emission source [66]. The ratio value of the petrogenic source was less than 0.2, while that of the mixed source was between 0.2 and 0.35. The contribution of combustion activity was demonstrated by a ratio larger than 0.35. The petrogenic source denominated five samples of the exposed school based on the analysis. In contrast, the mixed source may emit two samples, and the remainder of the samples (70.8%) may be released from the combustion activity.

Two vital anthropogenic sources contributed to air emissions in traffic, gasoline and diesel, comprising multiple constituents with congeners of high-molecular-weight PAHs. According to Fang et al., it is possible to use a diagnostic ratio formulated with FLU/(FLU + PYR) to discriminate between two origins of traffic emissions having a value less than 0.5, suggesting the origin of gasoline [67]. On the other hand, a value greater than 0.5 means a source is diesel-combusted. Diesel emissions (93.3%, *n* = 14) were the major influence for the comparative schools, while 100% samples of schools S2 and S3 were suspected of containing a gasoline marker, which may be contributed positively by heavy-duty vehicles passing by. Comparative schools are situated less than 50 m from the main road frequently used by palm oil lorries owned by the nearby plant oil plantation.

The exposed schools had a higher percentage of gasoline emissions, primarily due to heavy-duty vehicles (i.e., diesel-powered trucks) from the midstream area of the logistics hub of the petrochemical industry, contributing to the FLU/(FLU + PYR) diagnostic ratio. Gasoline combustion was also abundantly present in the samples of schools S1 and S2. The schools were less than 100 m from the main road and had a higher fraction of PAHs with a heavy molecular weight. This result was parallel to [5,42,68], as light- and medium-duty tailpipe vehicles contribute to a higher concentration of heavy congener PAH emissions. Lin et al. suggested that automobiles powered by gasoline contribute to PAHs of medium and high sub-atomic weight [69].

Furthermore, diesel-fuelled vehicles have also been correlated with lower PAH percentages. This study’s findings suggested that PAH sources may come from emissions from the oil and gas industry and traffic, wood, grass or coal combustion. A similar finding was reported in a previous study [11], as the location near a petrochemical complex was predominantly sourced from coal combustion based on the BaA/(BaA + CYR) diagnostic ratio measured at 0.67. In addition, it was proposed that the bituminous coal used in the petrochemical complex, in particular PYR, BaP and BbF, could emit a higher proportion of four- and five-ring PAHs.

### 3.3. Principal Components Analysis (PCA)

Three significant major principal components (PCs) for PAH emissions with a value above 1 were discovered by PCA, accounting for 75.06% variability data of the exposed schools. The contributing emission factors from the PCA of 28 samples of exposed schools and 18 samples of comparative schools are shown in Table 2. The hefty contribution of NaP, FLU, PYR, BbF, CYR, IcP and BgP was defined by PC1, with a total variance of 45.12%. BbF, CYR, IcP and BgP were indicative species for vehicle exhaust emissions [42,70]. Some studies have identified the emission of coke ovens as being supported by NAP, BgP and IcP [71,72]. Petroleum combustion is indicated by a higher fraction of HMW species, especially BbF, IcP and BgP [73]. Meanwhile, 14.48% of the overall variation was clarified by PC2, primarily weighted by BkF, BaA, and FLA. As a tracer for diesel emissions, the three species in PC2 were identified [71]. PC3 had a difference of 15.10% and had heavy ANT loading and moderate PHE and DbA loading. The superiority of ANT, PHE and DbA [74] was represented by wood combustion behaviour. Vehicular traffic and industrial pollution may be positively attributed to the mixture of PC1 and PC2.

On the other hand, three PCs were greatly derived from the same study, with 83.66% variance in the dataset of the comparative schools. With 47.04% of the overall variation, the first identified PC clarified the dominant contribution of six species (PHE, PYR, BbF, BaP, IcP and DbA). Meanwhile, with 27.27% data variance, the high loadings of BaA, BgP and CYR distinguished PC2. PC3 accounted for 9.34% of the overall variation, with heavy ANT loading and poor NAP and FLU loading. The activities of vehicle emissions (PC1), diesel combustion (PC2) and wood combustion (PC3) were better represented in comparative schools by the profile of PAHs.

### 3.4. Health Risk Assessment

Generally, exposed school S1 showed the highest TEQ values (20.01 ng m^−3^), followed by S3 (16.89 ng m^−3^) and S2 (13.69 ng m^−3^) (Appendix A). The TEQ values for comparative schools C1, C2, and C3 were 5.51, 0.54 and 0.21 ng m^−3^, respectively. A similar finding was observed in two elementary schools in Portugal, with TEQ values reaching up to 22.00 ng m^−3^ [75]. Besides, it was found that TEQ values in this study were higher than in studies reported in the urban city of Kuala Lumpur, i.e., 640.01 and 266.27 pg m^−3^, respectively [5,76]. Oliveira et al. mentioned that Asian schools had higher TEQ values (range: 4.70–49.4 ng m^−3^) than European schools (range: 0.04–29.8 ng m^−3^) [6].

On average, BaP contributed over 50% of TEQ values, thus explaining this congener’s predominant role in assessing the cancer risk due to PAH exposure [76]. Meanwhile, DbA portrayed the second-highest contributor to TEQ values ranging between 0.93% and 33.86%. Furthermore, the DbA congener predominantly contributed up to 69% for school C3, and the rest of the schools had an average percentage TEF of less than 50%. Oliveira et al. also identified that BaP and DbA were the most influential contributors to TEQ values in the health risk assessment [59].

Based on the ILCR model proposed by USEPA, the cancer risk due to exposure to PAHs through inhalation was determined [48]. The exposed children relatively had a substantially higher cancer risk than the comparative groups. It can be arranged in increasing order: C3 (1.37 × 10^−8^) < C2 (3.62 × 10^−8^) < C1 (3.81 × 10^−7^) < S2 (9.28 × 10^−7^) < S3 (1.09 × 10^−6^) < S1 (1.29 × 10^−6^). Differences in body weight, physiology and variation in PAH exposure in individuals contribute to the uncertainty in evaluating health risks [47]. Hence, a more precise cancer risk estimation of PAH inhalation was obtained through Monte Carlo simulation, as depicted in Figure 3. The mean value of the simulated probability cancer risk for the exposed and comparative schools was 1.20 × 10^−6^ and 8.38 × 10^−8^, respectively. Both mean values show similarity to the actual generated ILCR in SPSS software (1.13 × 10^−6^ and 1.6 × 10^−7^). The 95th percentiles of the ILCR calculated the risk for the exposed and comparative populations as 2.22 × 10^−6^ and 2.95 × 10^−7^, respectively.

The ILCR for the exposed children exceeds the acceptable risk limit of the USEPA reference and denotes a non-negligible risk [48]. The estimated carcinogen risk in this study for the exposed group was notably higher than two recent local studies in Kuala Lumpur (the year 2014: 3.32 × 10^−8^; the year 2019: 2.64 × 10^−8^), which was predominantly contributed by traffic emission [5,76]. Meanwhile, the magnitude of the ILCR for comparative schools was relatively lower than the risk calculated in elementary school children in Portugal, between 1.0 × 10^−8^ to 5.0 × 10^−8^ [75]. Oliveira et al. also eloquently concluded that Asian children are exposed to a higher magnitude of the ILCR (1.3 × 10^−6^ to 5.4 × 10^−5^) as compared to European children (5.9 × 10^−9^ to 1.1 × 10^−8^) [6]. Moreover, exposure to PAHs originating from vehicles, biomass, coal combustion and petroleum combustion increased the ILCR values to 2.1 × 10^−5^ for children in Ulaanbaatar City, Mongolia [77]. Drastic environmental health control measures are required to effectively mitigate the non-negligible cancer risks due to PAH exposure [78]. 

Sensitivity analysis disclosed the most persuasive variable for the carcinogenic risk due to PAH inhalation. For the exposed group, the children’s body weight (−0.78) and the PAH concentration in the TEQ value (0.42) were the most influential determinants in the ILCR estimation (Figure 4A). A negative value of correlation shows that the increase in the predictor is related to a decrease in the ILCR prediction. Meanwhile, the TEQ value in the comparative dataset contributed about 66.7% variance of the risk output (Figure 4B). Similarly, the body weight of children in the comparative area had an antagonistic relationship (−0.19), which is consistent with health risk assessment studies [51,79,80].

### 3.5. Individual Factors in DNA Damage

The tail moment of the exposed group (27.20 ± 8.21) was significantly longer than the value recorded in comparative school children (21.03 ± 4.88). The respondents from school S2 had the highest amount of DNA damage (31.89 ± 11.28) compared to children in other schools. On the other hand, the shortest tail moment, 20.43 ± 5.34, was shown by children in school C1. When statistically evaluated by one-way ANOVA, the result indicated that the exposed group had more substantial DNA damage than the comparative group (Appendix A). Interestingly, this result was higher than in previous studies [22,31], when it was found that children living in proximity to the petrochemical industry in Mexico had an olive moment of 9.52 (8.65, 10.48) and 8.3 (3.1–16.8), respectively. This disparity may be attributed to the different cellular samples used (lymphocytes vs. buccal epithelial cells) and the broader age span of children (6 to 12 years old).

Using independent *t*-test and one-way ANOVA, the individual tail moment variables were stratified and compared. The most significant factors for the comparative group were the age factor and mosquito coil use, according to Table 3. Compared to children aged 10 and 11 years, children aged 9 years had the shortest comet tail (exposed group: 25.40 ± 4.12; comparative group: 18.44 ± 3.38). In comparison, there was a significantly larger tail moment (27.24 ± 7.34) for children who lived with smoking family members than for children who were not exposed to cigarette smoke. In comparison, there was a considerably longer tail moment (27.41 ± 7.33) for grilled food users than for children who ate grilled food less regularly (27.08 ± 6.16). Interestingly, children who were exposed to mosquito coil pollutants, especially in the comparative community (22.27 ± 4.32 vs. 20.20 ± 4.20) (*p* < 0.05), had a slightly higher tail moment value.

Confounding factors such as exposure to tobacco smoke and age can interfere with the sensitive comet assay outcome [81]. A recent research by Aksu et al., for instance, found that DNA damage was more serious among smoking adults than in the control group, although no significant comparison was observed [82]. Cigarette smoke exposure among children has been seen to have a significant elevation in DNA damage [83,84]. Gajski et al. did not find any major effect of age on DNA damage among children, although it was found that gender had a statistical impact on the tail length, tail intensity and tail moment [85]. In comparison, 25 female children (50%) were observed to have significantly higher comet parameters than the male ones [85]. Recent research found that a group of male children living near a waste incinerator had more severe DNA damage than female children with a high heavy metal exposure burden [86].

The body mass index (BMI) did not influence the outcome of DNA damage in this current research. Overweight and underweight children, however, had a more extended tail moment than typical-BMI children. The finding agreed with a previous study [87], which confidently showed that a higher BMI among the population of healthy Swedish youngsters resulted in more severe DNA damage. As a result of the formation of reactive oxygen species and lipid peroxidation, excessive body fat has been postulated to impede DNA stability [88]. Regarding the intake of grilled foods, it was confirmed that they contain carcinogenic substances (i.e., PAHs) [89] and that they are associated with altering the structure of DNA adducts and may cause DNA damage [90]. Meanwhile, intake of fruit, vegetables and health supplements could suppress DNA damage [91]. The deficiency of micronutrients, such as vitamin C, vitamin E, zinc and antioxidants, could also escalate the risk of deformation of DNA strands and increase the chromosomal damage incidence [92].

Various scientific studies have shown the adverse impact of insect repellent and incense smoke exposure through animal and epidemiological studies [93,94,95]. Liu et al. found that a large concentration of volatile organic compounds (VOCs), PAHs, aldehydes and fine particles, which pose an acute and chronic health risk, could be generated by mosquito coils [96]. Besides, mosquito coils are the most toxic insect repellents than mat and liquid repellents [93] due to a higher lipid peroxidation response and increased free-radical substances that can alter the cell membrane and cause damage to DNA [95]. In this current study, DNA damage in buccal epithelial cells, especially among the comparative population, was significantly affected by mosquito coil exposure. An in vitro study by Szeto et al. confirmed the finding, as they discovered that insect repellents (i.e., incense burning) could cause degradation of the DNA strands in human lymphocytes [97]. The indoor air toxins of mosquito coils were also associated with lung cancer incidence [98,99].

### 3.6. Relationship between Tail Moment with PAH Exposure and Other Risk Factors

Individual factor and environmental PAHs exposure were assessed individually on how diverse the factors contributed to DNA damage among children. Simple linear regression demonstrated a highly significant relationship between PAHs concentration and DNA damage. Furthermore, the analysis also revealed a significant association between age, BMI, and open burning on DNA damage (Appendix A). Multiple linear regression was further performed to determine the best measure of the dependent variables in order to boost statistical aid. The statistical equations indicated that the tail moment increased with each increased unit of total outdoor PAHs and carcinogens, a decreased unit of non-carcinogenic PAHs and frequent open burning (Table 4). The first prediction model consists of two important variables (i.e., total outdoor PAHs and open burning), with F (3, 206) = 9.643, *p* < 0.0001 and adjusted R^2^ = 0.110, which significantly affect the tail moment. A combination of two variables suggested that 11.0% of the variability of the tail moment after confounder adjustment was clarified by the model. Similarly, the second model includes two significant variables (outdoor PAH concentration carcinogen and open burning), with F (4, 223) = 9.039, *p* < 0.0001 and adjusted R^2^ = 0.124; 12.4% of the tail moment variance was described. Due to the violation of the *p*-value, non-carcinogenic outdoor PAHs, age, BMI and mosquito coil were excluded from the model.

The significant variables for estimating the tail moment of F (5, 204) = 7.072, *p* < 0.0001 and adjusted R^2^ = 0.148 were the total concentration of indoor PAHs and open burning. After controlling all the possible confounders, the third model described approximately 14.8% of the variance of the tail moment. The fourth model, containing the concentration of carcinogenic indoor PAHs and open burning, which substantially predicted DNA damage in the buccal epithelial cell, with adjusted R^2^ = 0.127, was also obtained from the multivariate analysis. In other words, by the interaction of indoor PAH emissions and open burning, this model substantially explained a 12.7% variation of the tail moment. This result was in agreement with Gamboa et al., in which more severe DNA damage was found among Mexican children living close to the industrial area of oil extraction and correlated with chronic PAH pollution exposure [100]. In that analysis, one of the carcinogenic PAH species, BkF, was reported as significantly higher (47.29 ± 10.65 vs. 17.36 ± 3.79 ng m^−3^) relative to regions without oil extraction activity.

Jasso-Pineda et al. also revealed more significant DNA damage in a population of children living in a household that used biomass combustion and was strongly associated with internal PAH exposure, specifically urinary 1-hydroxypyrene (1-OHP) [20]. A stratified analysis by Sachez-Guerra et al. showed that excretion metabolite PAHs and DNA damage were significantly correlated, with DNA damage among children living near the petrochemical industry [31]. Likewise, Ruchirawat et al. eloquently illustrated a higher internal dose magnitude of PAH metabolites and DNA damage in urban children than in the comparative group [23]. In addition, a cross-sectional analysis in Kuala Lumpur, Malaysia, also showed a high risk of DNA injury and respiratory symptoms among children attending school near heavy traffic areas [101].

The common practice of domestic open burning of household waste was proposed to justify the higher incidence of respiratory disease in rural communities [102]. This research found that the practice of open burning is also statistically established as one of the causal factors for the higher occurrence of genotoxicity among children. Besides, it has been conclusively proven that a higher concentration of carcinogenic PAHs is generated by domestic open burning, especially during the dry season [103]. During school hours, the residential areas near the schools often burned waste, having a detrimental effect on air sampling. Therefore, public health issues will be caused by the illegal open-burning practice among the nearby population, especially during school hours. This leads to an unhealthy environment and may increase DNA damage among children.

In vitro studies have revealed synergism, additive or antagonism findings through the genotoxic potency of the PAH mixture [104,105,106]. The cytotoxicity and genotoxicity of a mixture of PAHs on HepG2 cells were researched in the year 2018 and disclosed an interesting finding [107]. It was found that the binary combination of carcinogenic BaP with non-carcinogen PAHs is cytotoxic; however, they do not have a genotoxic effect on the cells. The authors measured a decrease of 60% in chromosomal damage relative to a single BaP dose. The combination effect of BaP with BbF and BaA pairwise PAHs on the metabolic p53 pathway is synergistic [108]. Therefore, simultaneous inhalation of various congeners of PAHs has aggravating mechanistic effects on human health.

## 4. Conclusions

This present research concluded that sensitive receptors experience higher exposure to particulate PAHs in the exposed area. The results revealed that the highest concentration of total PAHs was reported at exposed school S2, with a value of 67.72 ± 49.84 ng m^−3^, followed closely by school S1 (64.64 ± 44.85 ng m^−3^) and school S3 (61.60 ± 39.74 ng m^−3^). On the other hand, low PAH concentrations were recorded in comparative schools, especially in schools C2 (5.93 ± 0.59 ng m^−3^) and C3 (6.36 ± 1.19 ng m^−3^). The source diagnostic ratio and PCA analysis indicated that the source of PAHs could come from the petrochemical industry, traffic emission and also wood combustion. Monte Carlo simulation predicted that the 95th percentiles of the ILCR for the exposed and comparative populations were 2.22 × 10^−6^ and 2.95 × 10^−7^, respectively. Generally, the cancer risk due to PAH inhalation for the exposed children exceeds the acceptable risk limit of the USEPA reference and denotes a non-negligible risk. Sensitivity analysis disclosed the body weight of children and PAH concentrations in the air were the most persuasive variables to estimate a precise carcinogenic risk due to PAH exposure.

Comet assay microscopy analysis found that the exposed groups had a significantly higher tail moment than the comparative groups. The mean tail moment values were 27.20 ± 8.21 and 21.03 ± 4.88 for the exposed and comparative groups, respectively. Stratified analysis revealed that the age factor and exposure to mosquito coils greatly affect the tail moment. This research strongly suggested that DNA damage is significantly affected by particulate PAHs after controlling all possible confounding factors (e.g., demographic, socio-economic, lifestyle and tobacco smoke exposure) in both study design and statistical analysis. The result provided evidence that children living in close proximity to the industrial zone could be subject to greater levels of exposure to environmental PAHs and a higher risk of genotoxicity than children living in less polluted areas.

Moreover, as limited epidemiological studies have been performed on the relationship between industrial air pollution and genotoxicity among children in Southeast Asia, the knowledge gap has been successfully reduced. New genotoxicity models have been successfully established, which forecast exposure to PAHs as a valuable health impact assessment (HIA) tool. The findings will help understand the levels, distribution and sources of PAHs in educational institutions, providing insights into the governance of the living environment and children’s well-being, particularly in the industrial area.

## Figures and Tables

**Figure 1 ijerph-18-02575-f001:**
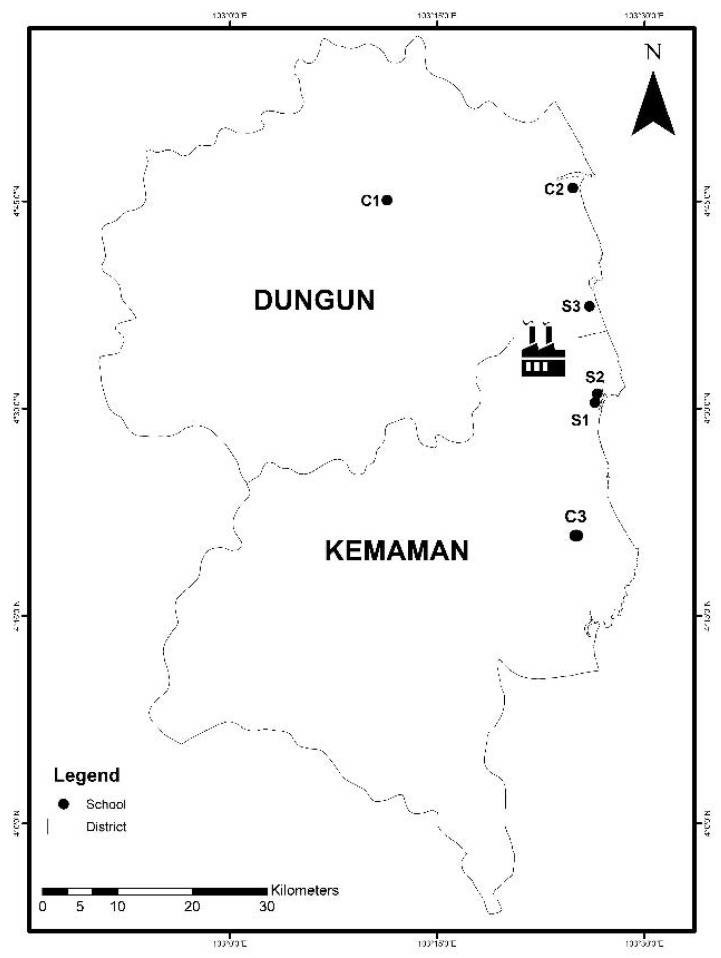
Map of peninsular Malaysia and the study area (Terengganu State). The six schools are located in the districts of Dungun and Kemaman, Terengganu.

**Figure 2 ijerph-18-02575-f002:**
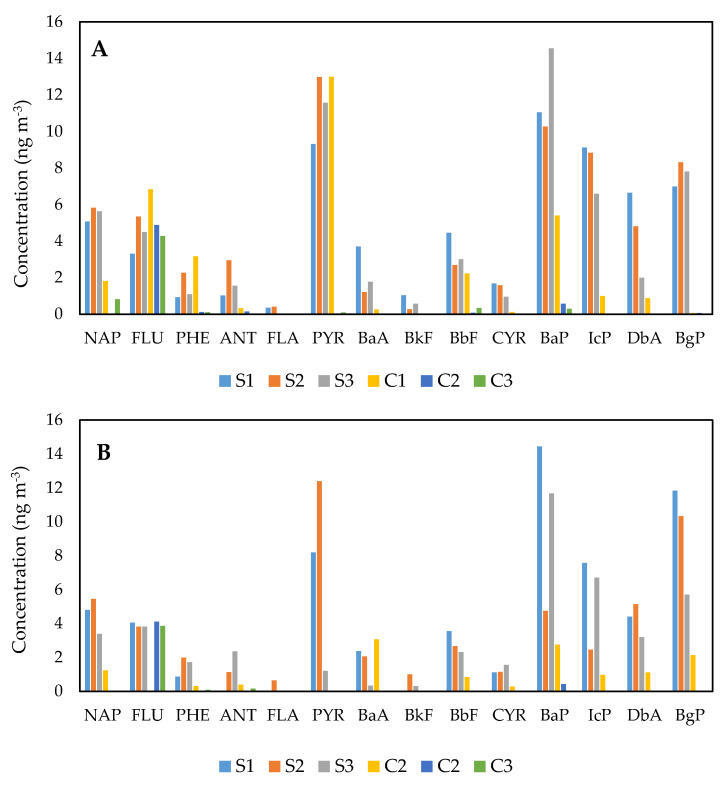
Distribution of polycyclic aromatic hydrocarbon (PAH) species in PM_2.5_ samples ((**A**): outdoor; (**B**): indoor).

**Figure 3 ijerph-18-02575-f003:**
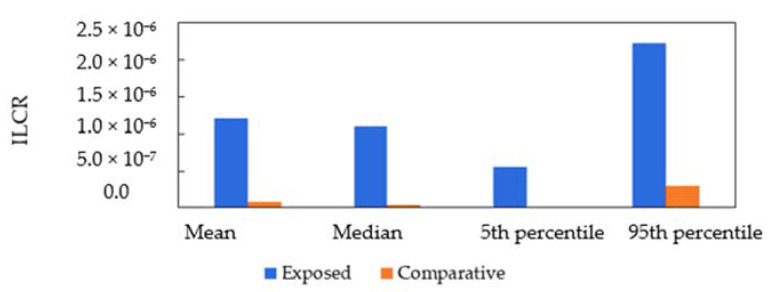
Probabilistic distribution of incremental lifetime cancer risk (ILCR) for exposed and comparative children.

**Figure 4 ijerph-18-02575-f004:**
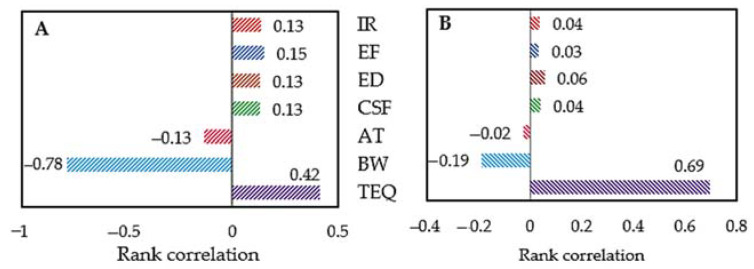
Sensitivity analysis of carcinogenic risk for exposed children (**A**) and comparative children (**B**).

**Table 1 ijerph-18-02575-t001:** Types of variable distribution used in the Monte Carlo simulation.

Variable	Unit	Distribution Mode	Exposed School	Comparative School
Toxicity equivalent concentration (TEQ)	ng m^−3^	Logistic & Log-normal	17.35	2.21, 2.51
Inhalation rate (IR)	m^3^ day^−1^	Constant	12	12
Exposure frequency (EF)	day year^−1^	Constant	250	250
Exposure duration (ED)	year	Constant	6	6
Averaging time	days	Constant	25,500	25,500
Body weight (BW)	kg	Negative binomial	0.206	0.307
Cancer slope factor (CSF)	mg kg^−1^ day^−1^	Constant	3.85	3.85

The value for logistic data is an arithmetic mean, the value for log-normal distribution is LN (arithmetic mean, standard deviation) and the negative-binomial distribution data are represented by a *p*-value.

**Table 2 ijerph-18-02575-t002:** Principal components analysis with varimax rotation.

Principal Component (PC)	Species	Factor Loading	Eigenvalue	Variability (%)	Source
PC1 Exposed schools	NAP	0.824	7.201	45.120	Vehicle and coke oven
	FLU	0.829			
	PYR	0.741			
	BbF	0.827			
	CYR	0.859			
	IcP	0.889			
	BgP	0.804			
	BaP	0.630			
PC2 Exposed schools	BkF	0.828	1.782	14.836	Gasoline
	FLA	0.728			
	BaA	0.758			
	BbF	0.293			
	BaP	0.233			
PC3 Exposed schools	ANT	0.948	1.526	15.103	Wood combustion, diesel
	PHE	0.700			
	DbA	0.558			
	NAP	0.341			
	PYR	0.335			
PC1 Comparative schools	PHE	0.853	7.130	47.040	Vehicle
	PYR	0.931			
	BbF	0.938			
	BaP	0.907			
	IcP	0.918			
	BgP	0.832			
PC2 Comparative schools	BaA	0.941	2.857	27.27	Diesel
	BgP	0.954			
	CYR	0.714			
PC3 Comparative schools	ANT	0.977	1.080	9.34	Wood combustion
	NAP	0.231			
	FLU	0.258			

**Table 3 ijerph-18-02575-t003:** Comparison of individual factors in the tail moment among children.

Variables	Exposed Group (*n* = 85)		Comparative Group (120)	
	Mean ± SD	*p*-Value	Mean ± SD	*p*-Value
**Age (year)**				
**9**	25.40 ± 4.12	0.409	18.44 ± 3.38	0.005 *
**10**	26.70 ± 6.07		21.98 ± 4.50	
**11**	27.96 ± 7.38		20.46 ± 4.08	
**Exposure to tobacco smoke**				
**Yes**	27.24 ± 7.34	0.915	20.86 ± 474	0.674
**No**	27.08 ± 6.31		20.52 ± 3.91	
**Grilled food**				
**Yes**	27.41 ± 7.33	0.804	21.14 ± 4.49	0.354
**No**	27.08 ± 6.16		20.37 ± 4.17	
**Supplement**				
**Yes**	26.98 ± 6.76	0.646	20.53 ± 4.05	0.584
**No**	27.60 ± 6.56		21.06 ± 4.92	
**Mosquito coil**				
**Yes**	30.05 ± 7.11	0.085	22.27 ± 4.32	0.034 *
**No**	26.71 ± 6.46		20.20 ± 4.20	
**Open burning**				
**Yes**	26.15 ± 6.33	0.350	20.56 ± 4.40	0.766
**No**	27.53 ± 6.73		20.79 ± 4.23	

* *p*-Value is significant at level 0.001.

**Table 4 ijerph-18-02575-t004:** Prediction models of the tail moment after controlling all confounders.

Model	Adj R^2^
Model 1 Tail moment = 12.892 + 0.054 (total outdoor PAHs)—2.415 (open burning)	0.110
Model 2 Tail moment = 14.120 + 0.170 (carcinogen outdoor PAHs)—1.870 (open burning)	0.124
Model 3 Tail moment = 13.345 + 0.076 (total indoor PAHs)—2.190 (open burning)	0.115
Model 4 Tail moment = 15.468 + 0.187 (carcinogen indoor PAHs)—2.328 (open burning)	0.127

## Data Availability

Not applicable.

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
