# Peer review of "Exposure to Particulate PAHs on Potential Genotoxicity and Cancer Risk among School Children Living Near the Petrochemical Industry"

_ijerph, 2021, doi:10.3390/ijerph18052575_

Round 1

Reviewer 1 Report

This study assessed PM2.5 PAHs exposure on potential genotoxicity and cancer risk among children near the petrochemical industry and community in Malaysia. Daily average PM2.5 PAHs were analyzed at six primary schools near industrial areas and three comparative schools far away from any industrial activity.  Comet Assay was utilized to assess the DNA damage for selected 205 children.. PAHs in the exposed schools are mostly from the vehicle and industrial emissions and the study showed that higher exposure to PAHs had increased the risk of genotoxic effects and cancer risk among the children. Several comments were suggested before publication.

  1.       How many filter samples were collected in total for PAH analysis in the study and what was the frequency of sampling?
  2.       How many samples were used for PCA analysis for exposed and comparative groups in section 3.3?

3.       In section 3.6, why only two variables were included in the prediction models of tail moment? Why choose open burning instead of mosquito coil which had a higher value of the tail moment? Any explanation on the regression coefficients from the multiple linear regression, especially the negative values for open burning?

Author Response

Thanks a lot for your helpful comments. We have revised our paper accordingly and feel that the reviewer’s comments helped clarify and improve this manuscript. Please find our response in red font colour.

Point 1: what was the frequency of sampling?

Response 1: The 24-hour air sampling for five consecutive school days was carried out at three primary schools located within 5 km from the petrochemical industrial area and was referred to as exposed school S1, S2, and S3. Meanwhile, C1, C2, and C3 were comparative schools in three schools situated more than 20 km from the industrial area and away from any industrial activity. Both the exposed and the comparative regions were situated in Terengganu, Malaysia (Figure 1). The air samples were collected between January to May 2017.

Please refer to page 2, line 76 and 83.

Point 2: How many filter samples were collected in total for PAH analysis in the study

Response 2: A total of 60 indoor and outdoor PM2.5 were collected using low volume air samplers equipped with cyclone (Airmetrics Mini Vol) with a flow rate of 5 L min−1 through weighted quartz microfiber filters (47 mm diameter size). (Page 3, line 213-215)

Point 3: How many samples were used for PCA analysis for exposed and comparative groups in section 3.3?

Response 3: The contributing emission factors from the PCA analysis of 28 samples of exposed schools and 18 samples of comparative schools are shown in Table 2 (Page 9, Line 376-378)

Point 4: In section 3.6, why only two variables were included in the prediction models of tail moment? Why choose open burning instead of mosquito coil which had a higher value of the tail moment?

Response 4: The exposure of individual factors and environmental PAHs was individually evaluated on how diverse the factors among children contributed to DNA damage. Significant associations between the PAH concentrations, age, BMI and open burning on DNA damage was shown by simple linear regression (Table S6). (Page 13, line 516-519)

The non-carcinogen outdoor PAHs, age, BMI, and mosquito coil were excluded from the model due to the violation of the p-value. (Page 14, Line 530-531).

Point 5: Any explanation on the regression coefficients from the multiple linear regression, especially the negative values for open burning?

Response 5: Open burning is a categorical data. Negative value implies 'yes' for open burning. Please refer to Table S6.

Reviewer 2 Report

This is a valuable study that addresses the impact of air pollution, and particularly polycyclic aromatic hydrocarbons, on children’s health. The study will be of interest to the readers of IJERPH. This reviewer requests to authors to do a major revision that must be completed prior to acceptance for publication.

Major limitation of the study: Lack of sufficient information about Cancer Slope Factors and reference doses (RfD) for different polycyclic aromatic hydrocarbons. Numerical values of Cancer Slope Factors and RfDs may change overtime, and therefore cancer risk estimates and hazard quotients may also change. This should be acknowledged in the Discussion as an uncertainty and an area of ongoing research. While this uncertainty would not change the conclusions of the study about differences in health risks between industrial and non-industrial areas, it will have an effect on numerical estimates of risk.

Table S3 should be included in the Methods (not in the Supplementary Information), since the calculated risk values for cancer exposure depend on these numeric values. Units for the Cancer Slope Factor and the reference dose should be included in the table. This table should reference original studies that developed these slope factors and reference doses, and two references provided for the cancer slope factors are not sufficient, because those two publications did not develop the slope factors, but just adopted them from another literature source.

Grammatical and stylistic changes needed

Manuscript mentions tables A1, A2 and A3. Lines 152, 154: “The following Table A2 displays the numerical variables used in health risk assessment calculations for this study. Meanwhile, the CSF and RfD values involved in this study are portrayed in Table A3”. However, there are no tables A1, A2 and A3 in the manuscript.

Authors need to correct subject-verb agreement in the paper, so that subject and verb are either plural or singular.

Examples, from the Introduction

Line 32 Particle-bound polycyclic aromatic hydrocarbon (PAHs) is one… As authors indicate in the Keywords section, “PAHs” is plural noun; therefore, in the above sentence should use “are”, not “is”.

Second sentence in introduction, line 33, “It ubiquitously found in the environment and eventually enter the human body through three main routes of inhalation, ingestion, and dermal absorption”. Subject should be “They” referring to PAHs and sentence should start with “They are”.

Same correction for line 35.

Line 37, “PAHs congener” – should be “PAH congeners”.

This grammatical problem of subject-verb agreement is present in other sections in the manuscript and needs to be systematically addressed.

Other grammatical issue: lines 45-46, only one adverb should be used, in the sentence “effects on the children who live nearby the industrial areas have not been well investigated thoroughly.” Should be either “have not been well investigated” or “have not been investigated thoroughly”;

I recommend for the authors to work with a professional editor for copy-editing of the manuscript.

Author Response

Reviewer 2

Thanks a lot for your helpful comments. We have revised our paper accordingly and feel that the reviewer’s comments helped clarify and improve this manuscript. Please find our response in red font colour.

Point 1: Major limitation of the study: Lack of sufficient information about Cancer Slope Factors and reference doses (RfD) for different polycyclic aromatic hydrocarbons.

Response 1: The health risk assessment of PAHs inhalation was reassessed using exposure assessment, which refers to application of benzo(a)pyrene equivalent concentration (BaPeq) or also known as toxicity equivalent concentration (TEQ). Afterwards, the TEQ value was incorporated into the calculation of incremental lifetime cancer risk (ILCR) by using this equation:

where C indicates the carcinogenic PAHs based on BaPeq (ng m-3), IR is the air inhalation rate for children (12 m3 day-1) . EF stands for exposure frequency or total schooling day in the year 2017 (250 day year-1); ED denotes the exposure duration or length of time of contaminant contact for children (6 years). BW is the measured body weight of children during the sampling; AT is the averaging time of carcinogenic PAHs exposure (70 years x 365 days). CSF refers to inhalation carcinogenic slope for BaP, which is 3.85 mg kg-1 day-1. The information of variables used in the ILCR calculation was refered to USEPA and Peng et al (2011).

Please refer to section 2.5.

Point 2: Numerical values of Cancer Slope Factors and RfDs may change overtime, and therefore cancer risk estimates and hazard quotients may also change. This should be acknowledged in the Discussion as an uncertainty and an area of ongoing research. While this uncertainty would not change the conclusions of the study about differences in health risks between industrial and non-industrial areas, it will have an effect on numerical estimates of risk.

Response 2: In order to acknowledge the issue of uncertainty in health risk assessment, Monte Carlo simulation with 10,000 iterations were employed in this study to appraise the probabilistic ILCR range of children exposed to PAHs for the exposed schools and comparative area. Besides, a sensitivity analysis was computed to identify the most influential factor in determining the cancer risk among children. Please refer to line 167-179.

The ILCR model prediction could overestimate the actual carcinogenic risk or under-estimate it. Consequently, the outcome of cancer risk will not be applied to the entire population of interest. Integrating variability and uncertainty into the risk model is a more precise approach to determine probabilistic carcinogenic health risk. Therefore, the Monte Carlo simulation with 10,000 iterations was employed in this study to appraise the probabilistic ILCR range of children exposed to PAHs for the exposed schools and comparative area. The simulation also integrates the uncertainty and variability of the measured PAHs concentration and the specific determinant of children exposure [47, 50-51]. Afterwards, a sensitivity analysis was conducted to identify input parameters that significantly influence the estimated risk. Both analyses were implemented using the Crystal Ball software (version 11.1.2.4; Oracle Corp., USA).

Please refer to page 11, line 415-425 it mentions the discussion on uncertainty.

Based on the ILCR model proposed by USEPA, the cancer risk of exposure to PAHs through inhalation was determined [48]. The exposed children relatively had a substantially higher cancer risk than the comparative groups. It can be arranged in an increasing order: C3 (1.37-08) < C2 (3.62E-08) < C1 (3.81E-07) < S2 (9.28E-07) < S3 (1.09E-06) < S1 (1.29E-06). Individuals differed in body weight, physiological differences, and variation in PAHs exposure contribute to uncertainty in evaluating health risks [47]. Hence, a more precise cancer risk estimation of PAHs inhalation was obtained through Monte Carlo simulations, as depicted in Figure 3. The mean value of simulated probability cancer risk for the exposed and comparative groups was 1.20E-06 and 8.38E-08, respectively. Both mean values show similarity to the actual generated ILCR in the SPSS software (1.13E-06 and 1.6E-07). The 95th percentiles of ILCR calculated the risk for the exposed children and comparative populations was 2.22E-06 and 2.95E-07, respectively.

Point 3: Table S3 should be included in the Methods (not in the Supplementary Information), since the calculated risk values for cancer exposure depend on these numeric values. Units for the Cancer Slope Factor and the reference dose should be included in the table. This table should reference original studies that developed these slope factors and reference doses, and two references provided for the cancer slope factors are not sufficient, because those two publications did not develop the slope factors but just adopted them from another literature source.

Response 3:  Table 3 was omitted from the main text and new references was applied to calculate the ILCR. Please refer to page 4.

Point 4: Grammatical and stylistic changes needed

Response 4: A professional proofreader has proofread this manuscript.

Reviewer 3 Report

This work discusses the effects of particulate pollution on the health of children living in the vicinity of industrial areas. The study appears innovative as these effects appear to not be sufficiently studied by the existing literature. The work is well structured; however, I suggest the following revisions to improve some of its parts:

1) In the abstract, remove the number and the head of each sub-section that has been included. This must be a single section.

2) The following sentence is unclear "Children have been postulated as one of the most susceptible populations due to their physical and biological conditions, which are still under the developmental stage [15]". Perhaps it would be more appropriate to say that children are one of the most susceptible portions of the population due to their physical and biological conditions, which are still under the developmental stage.

3) The following statement is not very clear to me “The probabilistic risk assessment framework in determining the health risk will consider one assumption, similar to daily exposure to PAHs. In other words, the calculation of the semi-empirical equation will consider 365 days' exposure to PAHs. " Could you please explain better?

4) The authors should better explain how the PAHs concentration in air was calculated.

5) Are there potential aggravating effects related to the simultaneous inhalation of multiple types of PAHs? If these are known, please include some details.

6) Table 1 indicates FLR and FLP, these contaminants are not indicated in the manuscript and for them the reference dose values are not given in the supplementary material. Could you please integrate the necessary information into the text and supplementary material.

7) I suggest including some appropriate literature in the risk analysis section, for example the following is recommended:

- U.S. EPA. (2006). A Framework For Assessing Health Risk of Environmental Exposures To Children) - since it seems that the risk analysis was carried out in accordance with the USEPA.

- Atmospheric Pollution Research (2021), 12(2): 432-442; Process Safety and Environmental Protection (2020), 137: 223–237 - that are further similar works using the methodology for risk assessment due to atmospheric dispersion of pollutants

- Atmospheric Environment (2017), 160: 27-35 – this includes a risk assessment for PAH diffusion in the atmosphere

- Environmental Pollution (2021), 268, 1157723 – this includes a risk assessment for PAH also from other exposure pathways

Author Response

Thanks a lot for your helpful comments. We have revised our paper accordingly and feel that the reviewer’s comments helped clarify and improve this manuscript. Please find our response in red font color.

Point 1: In the abstract, remove the number and the head of each sub-section that has been included. This must be a single section.

Response 1: Abstract has been updated accordingly.

Point 2: The following sentence is unclear "Children have been postulated as one of the most susceptible populations due to their physical and biological conditions, which are still under the developmental stage [15]". Perhaps it would be more appropriate to say that children are one of the most susceptible portions of the population due to their physical and biological conditions, which are still under the developmental stage.

Response 2: The sentence has been revised (Line 47-48).

Point 3: The following statement is not very clear to me "The probabilistic risk assessment framework in determining the health risk will consider one assumption, similar to daily exposure to PAHs. In other words, the calculation of the semi-empirical equation will consider 365 days' exposure to PAHs. " Could you please explain better?

Response 3: The statement was omitted from the manuscript. It was replaced with:

Risk assessment is an essential method for assessing the adverse effects of PAHs exposure on human health, defining the target organ, and identifying the particular effects of chemicals to find an appropriate risk minimisation solution [43]. Carcinogenicity potency PAHs can be determined based on benzo(a)pyrene equivalent concentration (BaPeq) or also known as toxicity equivalent concentration (TEQ). To obtain the TEQ for the PAHs compounds, it requires the involvement of reference toxicity equivalent factors (TEFs) proposed by Nisbet and Lagoy [44], relative to the carcinogenic potency of BaP. The TEQ was determined based on a constant TEF value multiplied with individual PAHs' concentration, as indicated by the following equation.

Please refer to line 140-148.

Point 4: The authors should better explain how the PAHs concentration in air was calculated.

Response 4: The PAHs concentration was determined through an internal calibration standard containing a known concentration mixture of 16s PAHs congeners. Meanwhile, the final concentration PAHs in the air was calculated using this equation, where total air volume was 7.2 m3.

             C (ng m-3) = C determined (ng ml-1) x dilution factor/air volume (m3     

Point 5:  Are there potential aggravating effects related to the simultaneous inhalation of multiple types of PAHs? If these are known, please include some details.

Response 5: In vitro studies revealed synergism, additive, or antagonism findings through the genotoxic potency of the PAH mixture [105-107]. The cytotoxicity and genotoxicity of the mixture of PAHs on HepG2 cells were researched in the year 2018 disclosed an interesting finding [108]. It was found that the binary combination of carcinogenic BaP with non-carcinogen PAHs was cytotoxic; however, they did not induce a genotoxic effect on the cells. They measured a decrease of 60% of chromosomal damage relative to a single BaP dose. The combined effect of BaP with BbF and BaA pairwise PAHs on the p53 pathway is synergistic, and the health risk of these mixtures increases compared to that of the individual ones. Moreover, the combined effect of the mixture BaP with BbF and BaA on the metabolic p53 pathway is synergistic, and the health risk of these mixtures increases compared to a single PAH congener [109]. Therefore, simultaneous inhalation of various congener of PAHs imposes aggravating mechanistic effects on human health.

Please refer to page 14, last paragraph. line 1181-1188.

Point 6:  Table 1 indicates FLR and FLP, these contaminants are not indicated in the manuscript and for them the reference dose values are not given in the supplementary material. Could you please integrate the necessary information into the text and supplementary material.

Response 6: The term fluorene (FLU) and fluoranthene (FLA) has been standardized into the text and supplementary material.

7) I suggest including some appropriate literature in the risk analysis section, for example the following is recommended:

- U.S. EPA. (2006). A Framework For Assessing Health Risk of Environmental Exposures To Children) - since it seems that the risk analysis was carried out in accordance with the USEPA.

- Atmospheric Pollution Research (2021), 12(2): 432-442; Process Safety and Environmental Protection (2020), 137: 223–237 - that are further similar works using the methodology for risk assessment due to atmospheric dispersion of pollutants

- Atmospheric Environment (2017), 160: 27-35 – this includes a risk assessment for PAH diffusion in the atmosphere

- Environmental Pollution (2021), 268, 1157723 – this includes a risk assessment for PAH also from other exposure pathways

Response: Recent appropriate literature (year 2019-2021) has been added in the Methodology and Discussion for health risk assessment.

Please refer to section 2.5 and 3.4.

Round 2

Reviewer 2 Report

Manuscript has been revised appropriately.